# Consensus Definition and Prediction of Complexity in Transurethral Resection or Bladder Endoscopic Dissection of Bladder Tumours

**DOI:** 10.3390/cancers12103063

**Published:** 2020-10-20

**Authors:** Mathieu Roumiguié, Evanguelos Xylinas, Antonin Brisuda, Maximillian Burger, Hugh Mostafid, Marc Colombel, Marek Babjuk, Joan Palou Redorta, Fred Witjes, Bernard Malavaud

**Affiliations:** 1Department of Urology, Institut Universitaire du Cancer, 31059 Toulouse CEDEX 9, France; Roumiguie.Mathieu@iuct-oncopole.fr; 2Department of Urology, Hôpital Cochin, APHP, 75014 Paris, France; evanguelos.xylinas@aphp.fr; 3Department of Urology, 2nd Faculty of Medicine, Charles University, Teaching Hospital Motol, 15006 Prague, Czech Republic; antonin.brisuda@fnmotol.cz (A.B.); Marek.Babjuk@fnmotol.cz (M.B.); 4St. Josef, Klinik für Urologie, Caritas-Krankenhaus, 93053 Regensburg, Germany; mburger@caritasstjosef.de; 5Department of Urology, Royal Surrey County Hospital, Surrey, Guildford GU2 7RF, UK; hugh.mostafid@nhs.net; 6Department of Urology, Hôpital Edouard Herriot, 69437 Lyon, France; marc.colombel@chu-lyon.fr; 7Department of Urology, Fundacio Puigvert, 08025 Barcelona, Spain; jpalou@fundacio-puigvert.es; 8Department of Urology, Radboud UMC, 6525 GA Nijmegen, The Netherlands; Fred.Witjes@radboudumc.nl

**Keywords:** bladder cancer, transurethral resection, en-bloc resection

## Abstract

**Simple Summary:**

Transurethral resection of bladder tumours may be technically challenging. Complexity was defined by consensus from the literature by a panel of ten senior urologists as “any TURBT/En-bloc dissection that results in incomplete resection and/or prolonged surgery (>1 h) and/or significant (Clavien-Dindo ≥ 3) perioperative complications”. Patient and tumour’s characteristics that suggested to by the panel to relate to complex surgery were collected and then ranked by Delphi consensus. They were tested in the prediction of complexity in 150 clinical scenarios. After univariate and logistic regression analyses, significant characteristics were organized into a checklist that predicts complexity. Receiver operating characteristics (ROC) curves of the regression model and the corresponding calibration curve showed adequate discrimination (AUC = 0.916) and good calibration. The resulting Bladder Complexity Checklist can be used to deliver optimal preoperative information and personalise the organisation of surgery.

**Abstract:**

Ten senior urologists were interrogated to develop a predictive model based on factors from which they could anticipate complex transurethral resection of bladder tumours (TURBT). Complexity was defined by consensus. Panel members then used a five-point Likert scale to grade those factors that, in their opinion, drove complexity. Consensual factors were highlighted through two Delphi rounds. Respective contributions to complexity were quantitated by the median values of their scores. Multivariate analysis with complexity as a dependent variable tested their independence in clinical scenarios obtained by random allocation of the factors. The consensus definition of complexity was “any TURBT/En-bloc dissection that results in incomplete resection and/or prolonged surgery (>1 h) and/or significant (Clavien-Dindo ≥ 3) perioperative complications”. Logistic regression highlighted five domains as independent predictors: patient’s history, tumour number, location, and size and access to the bladder. Receiver operating characteristic (ROC) analysis confirmed good discrimination (AUC = 0.92). The sum of the scores of the five domains adjusted to their regression coefficients or Bladder Complexity Score yielded comparable performance (AUC = 0.91, C-statistics, *p* = 0.94) and good calibration. As a whole, preoperative factors identified by expert judgement were organized to quantitate the risk of a complex TURBT, a crucial requisite to personalise patient information, adapt human and technical resources to individual situations and address TURBT variability in clinical trials.

## 1. Introduction

Bladder cancer is the seventh most prevalent cancer worldwide [1] and the sixth leading cause of cancer in the EU, where it entails a significant burden in healthcare organization and cost [2]. Most patients present with non-muscle invasive bladder cancer (NMIBC), for which endoscopic resection or en-bloc dissection of bladder tumours, collectively referred to as transurethral resection of bladder tumour (TURBT), initiate the treatment and inform the risks of recurrence and progression. Pathology also provides information on the adequacy of surgery that is visually complete resection and presence of muscle at the resection base [3]. Although this is the most common procedure in oncologic urology, with over 120,000 new cases across Europe annually [2], few reports have addressed how individual characteristics may challenge the successful completion of surgery [4,5]. In addition, the reported variability of residual disease [6] and higher performances of experienced surgeons [7] emphasize the demands of “good-quality” TURBT [7]. Moreover, quality represents latent information for the non-expert, contrary to clinical complications that are self-evident, closely monitored by the public and insurers and used as proxy for quality metrics [8].

Any system capable to document how individual presentations influence surgical outcomes would be of high clinical relevance. Therefore, the objective of the present consensus was to detail and organize the factors based on which experienced urologists anticipate a complex TURBT.

## 2. Results

### 2.1. Step 1: Definition of Complexity

A PubMed search of “transurethral resection” (of) “bladder” and “morbidity” or “complication”, or “mortality” or “death” yielded 585, 664, 9 and 95 articles, respectively. Of these, 89 articles relevant to the process of defining complexity were analysed, obtaining 36 articles (Appendix A) which were instrumental in highlighting adequacy, operative time and morbidity as the three drivers that characterize a complex surgery, as opposed to an uneventful procedure [4,8,9,10,11,12,13,14,15,16,17,18,19,20,21,22,23,24,25,26,27,28,29,30,31,32,33,34,35,36,37,38,39,40,41,42].

After a single round of circulation, all panellists validated the following definition of a complex TURBT: “any TURBT/En-bloc dissection that results in incomplete resection and/or prolonged surgery (>1 h) and/or significant (Clavien-Dindo ≥ 3) perioperative complications”.

### 2.2. Step 2: Items That Drive Complexity

Eighty-five characteristics that were suggested by the panellists to influence surgery were organized into six chapters consistent with standard medical practice: patient’s characteristics and history, tumour characteristics, access to the bladder, bladder anatomy and surgical environment.

Their relevance was researched in two Delphi rounds, which showed consensus for 42 characteristics in the first round (Appendix A) and 83 in the second (Figure 1, Figure 2, Figure 3 and Figure 4). For any characteristic or item, the median opinion of the panel (Figure 1, Figure 2, Figure 3 and Figure 4) was then used as the metrics to weight its individual contribution to complexity.

### 2.3. Step 3: Construction, Discrimination and Accuracy of the Bladder Complexity Checklist Sum

#### 2.3.1. Clinical Scenarios

Smoking, underweight, normal weight and American Society of Anaesthesiologists (ASA) class 1–2 or 3 that in the panel’s opinions did not relate to the complexity of TURBT were not included in the scenarios, although age and sex that were also considered of little influence were retained, as they are standards in medical reporting. Although the surgical environment was consistently considered to have bearing on the odds of a complex surgery, the corresponding items were not included in the scenarios, as they were considered circumstantial rather than constitutive of the case. As a whole, 150 scenarios that included 9 items organized 5 five domains (Table 1) were presented to the panel. The members were strongly consistent in their anticipation of complexity, as consensus was observed for 131/150 (87.3%) scenarios that were by design confirmed for univariate and multivariate analysis. 

#### 2.3.2. Discrimination and Accuracy

In univariate analysis, the items that informed the tumour characteristics (number, location, size) and access to the bladder were significantly associated with complexity (Table 1). Patient’s history that did not reach statistical relevance still qualified for multivariate analysis (*p* = 0.07).

Five domains (Table 2) that in logistic regression were independent predictors of complexity, i.e., history, tumour number, location, and size and access to the bladder cavity, were used to develop the probability function that modelled the probability of a complex surgery.

(1)$$p(complex)\;=\;\frac{1}{1+\exp(13.34-0.99xHistory-0.96xTuNumber-1.44xMainTuLocation-1.04xMainTuSize-1.1xAccess)}$$

This function showed good discrimination (AUC: 0.92 (95%CI: 0.87–0.96) in receiver operating characteristic (ROC) analysis (Figure 5).

The simplification offered by the Bladder Complexity Checklist Sum (BCCS, Table 3) yielded comparable performance (C-Statistics *p* = 0.94, Figure 6).

Both instruments showed good calibration (Figure 3, Figure 4).

Figure 7 illustrates the balance between positive and negative predictive values according to increments in BCCS.

## 3. Discussion

Anticipation is essential to adapt staff and technical resources to individual challenges of clinical situations. The adoption of standardized instruments of evaluation for major urological procedures [43] spurred us to develop similar instruments for TURBT, the most common procedure in oncologic urology [2].

The first step contextualized complexity, a concept adapted to the rationalization of healthcare [44]. A PubMed search highlighted three dimensions that characterize a complex surgery, as opposed to a satisfactory and uneventful procedure. Adequacy was recently introduced in the European Association of Urology (EAU) guidelines to insist on the importance of complete resection of all visible tumours with the detrusor muscle in the specimen, a surrogate marker of resection quality that controls the risk of early recurrence [9] and may impact adjuvant treatment [11]. Surgery longer than one hour was included following a large population-based report from the American College of Surgeons National Surgical Quality Improvement Program (NSQIP), where it related to postoperative complications independently from age, comorbidities, tumour size and ASA classification [31]. Lastly, postoperative complications requiring surgical, endoscopic or radiological intervention—that is, Grade III and higher in the recently TURBT-adapted Clavien–Dindo classification [29]—were also considered, as they were recently shown [33] to affect a significant minority of patients (8.1%, of which 15% were Grade III and higher). Reminiscent of other major oncologic procedures (e.g., trifecta in kidney and prostate surgery), the consensus therefore encompassed the three reported qualifiers of complexity, oncological, procedural and postoperative into a multidimensional definition.

The second step researched robust clinical predictors. To that end, we relayed on expert judgement, a valuable instrument when other methods are intractable for scientific or practicable reasons [45]. TURBT appears to fall in that category, as although many factors are known to impact surgery and its outcomes [4,5,46], some important ones were not detailed in population-based series (e.g., position of the tumour) or were so infrequent as to elude detection (e.g., diverticulum). Conversely, experienced urologists are bound to encounter them along their career and to drive some operational conclusions as to the influence they may have on their management. This was confirmed by the extensive list of items drawn from experience and by the broad consensus of the panel on their relative contributions to complexity.

Most of the items that carried a “possibly”, “likely” or “very likely” risk of complication were consistent with the current literature. Conversely, some that had eluded cohorts [33] and population-based registries [4,31] made sense to the practising physician, notably, the access to the bladder cavity or the position of the tumours, with TURBT at the dome considered as “likely” to result in visually incomplete, lengthy or morbid surgery, compared to “very unlikely” for the trigon. The increments in scores with tumour sizes presented according to the current US procedural terminology (Figure 2) were in keeping with the increasing risks of complication and 30-day reoperation rates reported in two large NSQIP population-based studies [4,31]. A similar correlation was observed for the number of tumours, that is also a central parameter in the EAU/European Organisation for Research and Treatment of Cancer (EORTC) risk stratification of progression and recurrence [3].

Overall, high consistency between the literature or the practical constraints of surgery and the Delphi scores vindicated the present approach to anticipate complex TURBT.

However relevant, no single factor could possibly drive the entirety of the surgical challenge, which spurred us to the third step to analyse their respective contributions in random scenarios. Although the panel acknowledged the influence of technology in TURBT (Figure 4), elements pertaining to the surgical environment that were considered as adaptive rather than constitutive were not considered in the scenarios. Consistent with the format of clinical presentations, scenarios included age and sex, although they are considered of little bearing in TURBT (Figure 1). To account for the risk of cognitive overload [47], only four aspects were considered: patient’s history, tumour and bladder anatomy and access. Although this resulted in a high prevalence of complex cases (58/131 (44.2%) scenarios were classified as “possibly”, “likely” or “very likely” to result in incomplete resection or prolonged surgery (>1 h) or significant complications), random scenarios were preferred to collecting real-life clinical cases in the construction of the score, as this ensured that even rare situations were not overlooked.

On univariate analysis, tumour number, size, and location and access to the bladder cavity significantly related to complexity (Table 1). Although not significant in univariate analysis (*p* = 0.07), patient’s history still qualified for multivariate analysis, where all five aspects independently related to complexity.

As measured by their regression coefficients (Table 2), although patients’ history and bladder contributed to a lesser extent, tumour characteristics carried most of the information, thereby emphasizing the classical emphasis on thorough preoperative evaluation. The regression model showed excellent discrimination on ROC analysis (AUC: 0.92), while the calibration curve confirmed its accuracy (Figure 5).

The Bladder Complexity Checklist was then developed to facilitate the recording of significant characteristics in the clinic (Table 3). For illustration purposes, the case of a 75-year-old female patient with a thin bladder wall, showing a single 3 cm tumour of the dome would yield a sum of 15, consistent with a predictive value for complexity (PPV) of 100% (Figure 7). Summing the weight-adjusted scores of the Bladder Complexity Checklist carried similar discrimination and accuracy as the logistic model (Figure 4). This is to our knowledge the first effort to quantitatively inform with a simple clinical instrument the multidimensional complexity of TURBT. It could readily complement the other checklists proposed to control the quality of the procedure [37] or the step-by-step management of NMIBC [14].

Overall, the present methodology highlighted the factors that drove the anticipation by experienced surgeons of a complex TURBT. It would be amenable to other procedures where the surgical outcome relates to a large number of factors accessible to preoperative evaluation (e.g., radical prostatectomy, kidney transplantation). It also emphasized the variability in complexity of a procedure that is still widely regarded as menial.

The ability to anticipate and document complexity has important practical consequences. First, the Bladder Complexity Checklist could be instrumental in personalising the human and technical resources required for the most common procedure in oncologic urology [2]. This has become an absolute requisite in the current era of value-based care [48], where most procedural terminologies and reimbursement policies for TURBT consider the size and number of tumours compounded by comorbidity indexes, but overlook essential predictors such as the position of the tumour, a key descriptor of complexity in the present consensus. The Bladder Complexity Checklist Sum that organises and quantitates all relevant clinical information could also be used to drive the adaptation of health resources according to increments of complexity and support complexity-adapted coverage from health insurances.

Second, quantitating the difficulties entailed by a “good-quality” TURBT [7] would offer a solid ground to confront the morbidity and oncological outcome of a potentially complex procedure. Documenting variability is also important when analysing the benefits of different systems of resection or evaluating adjuvant treatments in research protocols [11]. Although all controlled trials to date overlooked the bias of complexity, we believe that crucial information such as the complexity score or, at the very least, a minimal dataset including size, number and position of the tumours should be documented and balanced in clinical research.

Third, measuring complexity that amounts to weighting the risks of the procedure would constitute an important instrument to inform the patient and therefore control part of his anxiety [49]. The constraints of information also include the training and experience of the surgical staff [50]. A large study from the NSQIP concluded that residents’ involvement in urology procedures was not associated with increased complications, although it significantly increased the operative time [27].

Regarding TURBT, the relation between time and complications [31] and surgeon experience and the presence of the detrusor muscle in the specimen [9] vindicated the panel’s prudent assessment of residents’ participation (Figure 4). This observation also has direct bearing on the organisation of care in academic hospitals, in terms not only of informed consent [50] but also of organizing the list so that cases showing high complexity receive proper attention in terms of consultant supervision and position on the surgical list [50].

Several limitations should be considered. First, it is recommended for health indicators to include panellists of different origins, from public health experts to patients’ representatives [51]. Here, the sole urologists’ perspective was adopted, which certainly contributed to the high degree of consensus and the strong consistency with clinicians’ experience. With 10 experts, the panel positioned at the first quartile of the distribution of panellists in a systematic review [51] of the Delphi methodology and was in line with the number of experts invited to develop other multidimensional instruments in urology [43].

Second, the model was not validated in the clinics, where a lower prevalence of complex cases may be anticipated. However, the review of 416 diagnostic studies showed that a lower prevalence improved specificity and had no systemic effect on sensitivity [52], suggesting that the current model would retain its relevance in the real-life setting. Third, important predictors such as the position or the multiplicity of tumours are best defined by preoperative flexible cystoscopy [53], which is optional when the diagnosis can be ascertained by medical imaging [3]. Last, the process yielded a large number of items (Table 3) that may require streamlining after the first returns of clinical experience.

## 4. Materials and Methods

The present Delphi method followed the recommendations of a systematic review for the development of healthcare quality indicators [51]. Six urologists designed the study into three separate work packages: definition of complexity, outline of the factors that drive complexity and evaluation of their respective contributions in clinical scenarios. Four panellists were then invited to broaden the scope of ages and experiences (Table 4). As a whole, the panel comprised 10 board-certified urologists with over 202 years of combined experience.

### 4.1. Step 1: Consensus Definition of Complexity

Reports on morbidity or mortality of TURBT were researched in the PubMed database (English language, 4/2009–4/2019, key words: “transurethral resection (of) bladder”, “morbidity”, “complication” “mortality” or “death”). A senior author (BM) reviewed all abstracts and analysed the articles of potential relevance before proposing to the panel a working definition of complexity in TURBT (Appendix A).

### 4.2. Step 2: Listing the Items That Drive Complexity

#### 4.2.1. Collection of the Factors Related to Complexity

Experts collected the factors that in their opinion could impact TURBT. All suggested items were considered and organized into domains, consistent with the medical usage and segmented according to the literature into a comprehensive list of items.

#### 4.2.2. Delphi Validation

The panellists scored the items using a five-point Likert scale, classifying from “very unlikely” to “very likely” the risk of complexity entailed by the individual items (Table 5). After the first Delphi round, they were informed of the panel’s distribution of the scores and requested in the second round to confirm or adjust their personal evaluation.

Consensus on an item was reached when the opinions across the panel were so consistent that the 95% confidence interval of their distribution was bounded within two consecutive scores. In subsequent analyses, the median value of the opinions or Median Opinion (MO) was used to weight the contribution of an item to complexity.

### 4.3. Step 3: Construction of the Bladder Complexity Checklist

#### 4.3.1. Construction of Clinical Scenarios

To acknowledge the multifactorial nature of complexity in medicine, items that reached consensus were then organized along clinical scenarios constructed by their random allocation within their respective domains of interest: patient’s history, tumour number, main tumour size, location, and structure, access to the bladder cavity. One hundred and fifty scenarios were constructed (Appendix A) and validated for clinical consistency (e.g., refuting the association of 30 mL prostate and female genital prolapse) by a senior author (B.M.). In keeping with the epidemiology of bladder cancer, twice as many scenarios were developed for male than female patients [54]. 

The panellists were requested to follow an adapted five-point Likert scale (Table 5) to answer the question: in the following scenario will TURBT result in incomplete resection or prolonged surgery (>1 h) or significant intra or postoperative complications (Clavien-Dindo Grade III and higher)?

Consensus was reached when the 95% confidence interval of the answers strictly showed “unlikely” as the upper bound (concluded as a scenario unlikely to be complex) or “possibly” as the lower bound (concluded as a possibly complex scenario). Otherwise, the answers were considered inconclusive, and the scenario was not considered for further analyses.

#### 4.3.2. Discrimination of Individual Items in the Prediction of Complexity

On univariate analysis, the two-tailed Mann–Whitney *U*-test tested in the 150 scenarios the relationship between the domains of interest and complexity, dichotomized as “very unlikely or unlikely” or “possibly, likely or very likely”.

Logistic regression was conducted, with the domains showing *p* < 0.1 on univariate analyses as predictors and complexity as a dependent variable. The probability of a complex surgery was estimated from the probability function. In keeping with the logistic regression model [55], it acknowledges the contributions of all independent domains (Table 2) by their respective regression coefficients adjusted to the specifics of the case by the median opinions of the panel (e.g., the respective contributions to complexity of a single tumour compared to 4 to10 tumours were 0.96 and 0.96 × 3, respectively, as shown in Figure 2).

Following the structure of the probability function:
(2)$$probability\;=\;\frac{1}{1\;+\;\exp(-x)}$$
where x is the sum of the intercept value of the logistic regression and of the scores of the independent domains multiplied by their regression coefficients, for any domain, the product of its regression coefficient by the score of its descriptor correlates with the probability of a complex surgery. This was used to simplify the function into a checklist (Table 3) where the respective inputs of the items were similarly quantitated by the product of the regression coefficient of their domains by the scores summarizing the median opinions of the panel (e.g., location on the anterior wall of the bladder; median opinion: 3 (Figure 2), regression coefficient of tumour location: 1.44 (Table 2), product: 3 × 1.44, approximated for ease of use to 4.5).

In any clinical situation, recording the most significant item in patient’s history and access to the bladder, in complement to the tumour number, main tumour location and size, calculated the Bladder Complexity Checklist Sum.

ROC curves of the model and of the Bladder Complexity Checklist Sum were compared by the C-statistics. Ultimately, calibration curves illustrated their accuracies in the estimation of the probability of complexity in individual scenarios [56].

STATA/MP was used for statistics (StataCorp, College Station, TX-USA), significance was set at *p* < 0.05.

## 5. Conclusions

Preoperative factors that relate to complex TURBT were identified by expert judgement and organized into the Bladder Complexity Checklist to facilitate the evaluation of the risk of a complex TURBT, a crucial requisite to personalise patient’s information, adapt human and technical resources to individual situations and address TURBT variability in clinical trials.

## Figures and Tables

**Figure 1 cancers-12-03063-f001:**
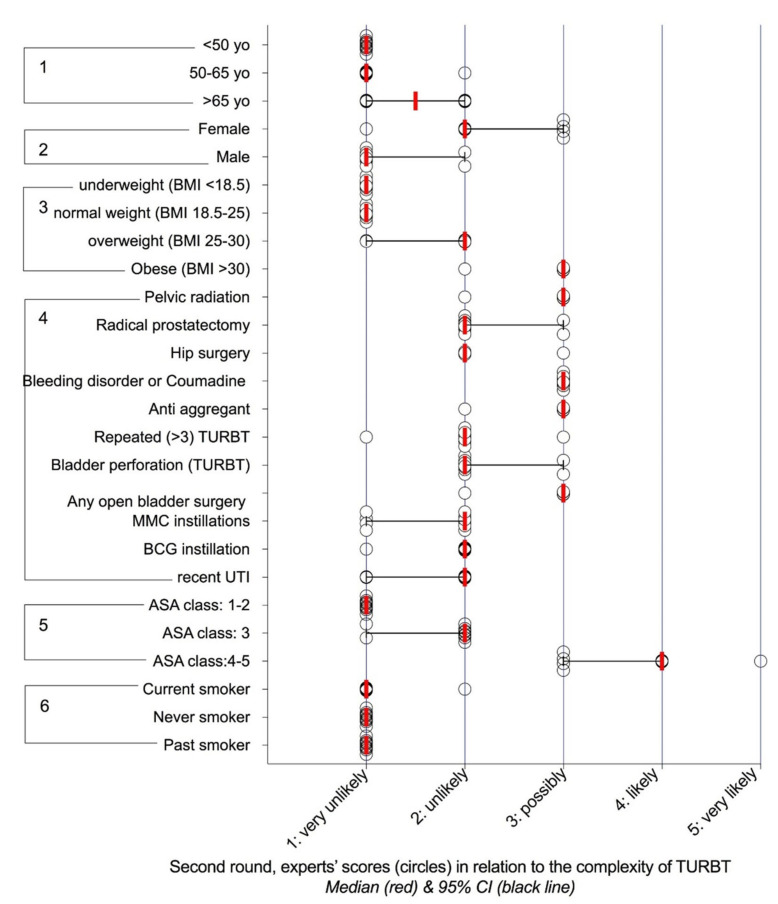
Distribution of the scores regarding the likelihood of incomplete resection and/or prolonged surgery (>1 h) and/or significant (Clavien-Dindo ≥ 3) perioperative complications according to patient’s characteristics. 1) age, 2) sex, 3) weight and body mass index (BMI), 4) patient’s history, 5) American Society of Anaesthesiologists’ (ASA) physical status classification, 6) tobacco smoking. MMC: Mitomycin C, Bacille Calmette Guérin (BCG), TURBT: transurethral resection of bladder tumour.

**Figure 2 cancers-12-03063-f002:**
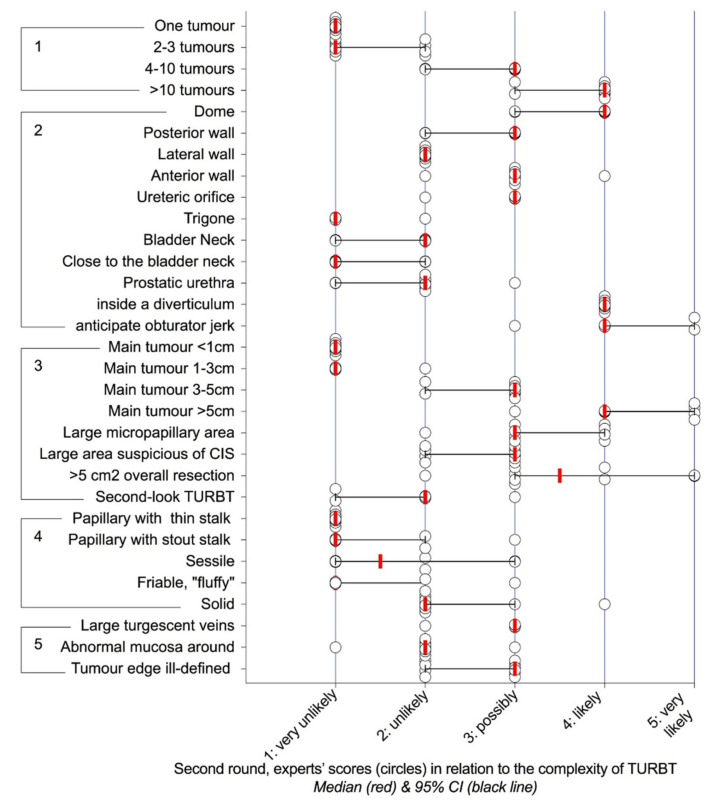
Distribution of the scores regarding the likelihood of incomplete resection and/or prolonged surgery (>1 h) and/or significant (Clavien-Dindo ≥ 3) perioperative complications according to tumour’s characteristics: 1) number, 2) location, 3) size, 4) structure, 5) surroundings. CIS: carcinoma in situ.

**Figure 3 cancers-12-03063-f003:**
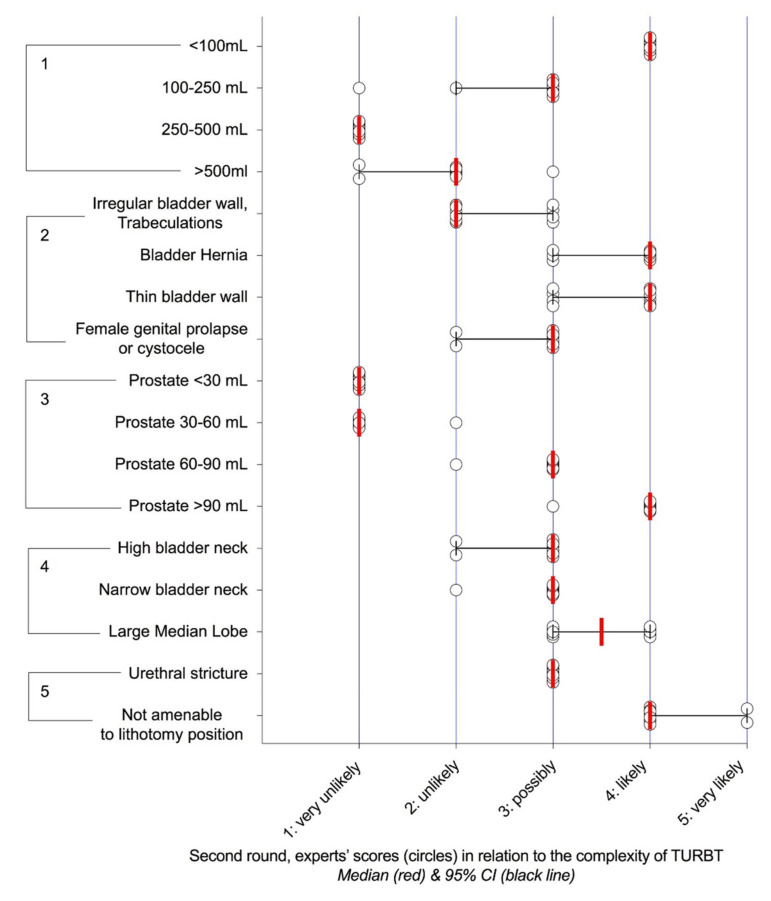
Distribution of the scores regarding the likelihood of incomplete resection and/or prolonged surgery (>1 h) and/or significant (Clavien-Dindo ≥ 3) perioperative complications according to bladder characteristics and access to the bladder cavity: 1) bladder capacity, 2) bladder structure, 3) prostate volume, 4) bladder neck, 5) others.

**Figure 4 cancers-12-03063-f004:**
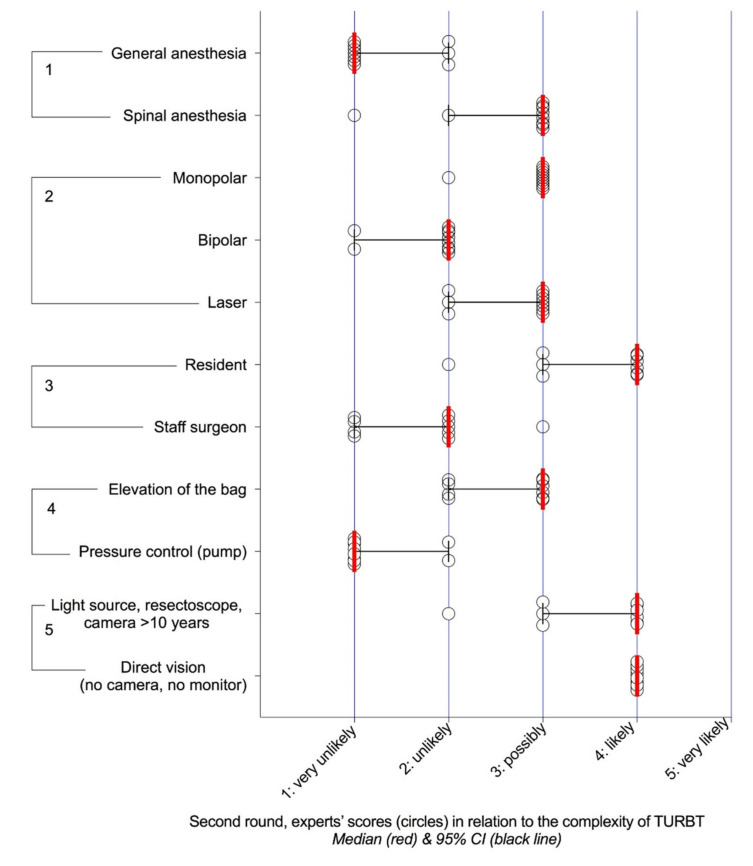
Distribution of the scores regarding the likelihood of incomplete resection and/or prolonged surgery (>1 h) and/or significant (Clavien-Dindo ≥ 3) perioperative complications according to the surgical environment: 1) anaesthesia, 2) energy, 3) operator, 4) bladder irrigation, 5) instruments.

**Figure 5 cancers-12-03063-f005:**
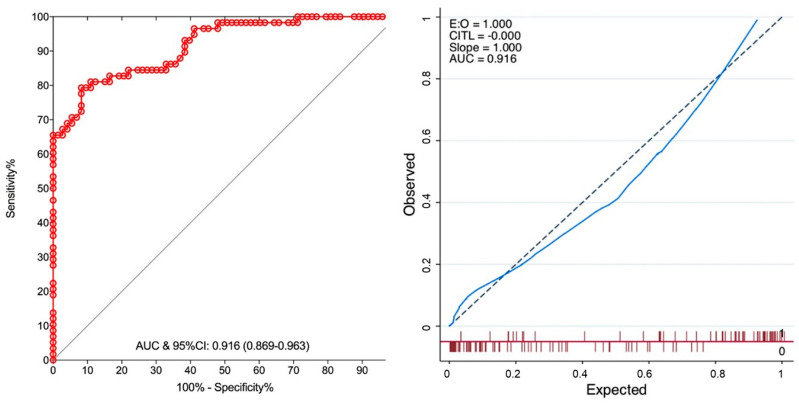
Receiver operating characteristics (ROC) curves of the regression model with the corresponding calibration curve showing adequate discrimination (AUC = 0.916) and good calibration, with calibration slope of 1 and calibration in the large (CITL) of 0, indicating that the predicted prevalence of complexity was in keeping with the observed prevalence (CITL) and that the model was not over fitted (slope).

**Figure 6 cancers-12-03063-f006:**
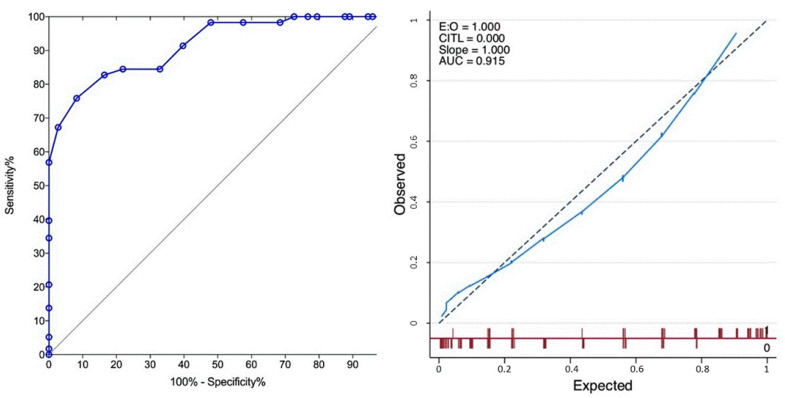
ROC curves of the Bladder Complexity Checklist Sum (BCCS) and the corresponding calibration curve showing similar discrimination and calibration performances compared to the regression model.

**Figure 7 cancers-12-03063-f007:**
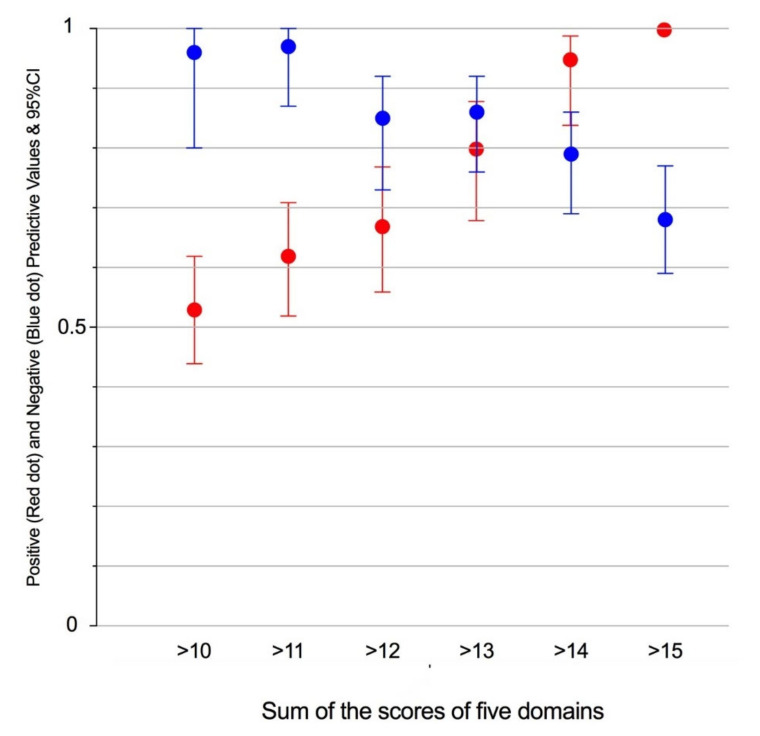
Negative (**blue**) and positive (**red**) predictive values (NPV and PPV) of increments in the BCCS.

**Table 1 cancers-12-03063-t001:** Univariate analysis of the scores of preoperative characteristics in a cohort of 131 random scenarios for which the panel was consistent in its anticipation of complexity.

Domain of Interest	Feature	Number of Items	Median Score, (95%CI)		Mann–Whitney *U*-Test
			TURBT Unlikely to Be Complex (*n* = 73)	TURBT Likely to Be Complex (*n* = 58)	
Patient’s characteristics	Age	3	1 (1–1)	1 (1–1)	n.s. (*p* = 0.85)
	Sex	2	1 (1–1)	1 (1–2)	n.s. (*p* = 0.72)
Patient’s history		12	1 (1–2)	2 (1–2)	n.s. (*p* = 0.07)
Tumour’s characteristics	Number	3	1 (1–1)	3 (3–4)	*p* = 0.002
	Location	10	3 (2–3)	4 (3–4)	*p* < 0.0001
	Size	5	2 (1–3)	3 (3–3)	*p* < 0.0001
	Structure	5	2 (2–3)	2 (2–3)	n.s. (*p* = 0.97)
Bladder Anatomy		8	3 (2–3)	3 (2–3)	n.s. (*p* = 0.82)
Access to the Bladder cavity		13	1 (1–3)	3 (3–4)	*p* < 0.0001

n.s. not significant.

**Table 2 cancers-12-03063-t002:** Logistic regression analysis showing independent relationships between the complexity of TURBT and patient history, tumour number, main tumour location and size and factors restraining the access to the bladder cavity.

Independent Variables	Regression Coefficient	Std. Error	z	*p* > |z|	95% CI of the Regression Coefficient	
Patient History	0.99	0.32	3.11	0.002	0.37	1.61
Tumour Number	0.96	0.23	4.18	0.000	0.51	1.41
Main Tumour Location	1.44	0.33	4.42	0.000	0.80	2.09
Main Tumour Size	1.04	0.26	3.98	0.000	0. 53	1.55
Access	1.10	0.26	4.31	0.000	0. 60	1.60
Intercept value	−13.34	2.31	−5.77	0.000	−17.87	−8.81

**Table 3 cancers-12-03063-t003:** Checklist detailing the five domains related to the prediction of a complex transurethral resection of bladder tumours by the panel. The Bladder Complexity Score (BCS) was calculated as the sum of the weight-adjusted scores. Increments in BCS relate to the positive and negative predictive values of experiencing a complex surgery, that is, “any TURBT/En-bloc dissection that results in incomplete resection and/or prolonged surgery (>1 h) and/or significant (Clavien-Dindo ≥ 3) perioperative complications”.

	Patient’s Characteristics		Tumour’s Characteristics		
Weight-Adjusted Scores	Medical History	Bladder Access	Number	Size	Location
1	No Relevant History	No relevant features	1–3	<3 cm	
1.5					Trigon
2	Hip Surgery Radical Prostatectomy Repeated TURBT (>3) Prior Bladder perforation MMC or BCG instillations UTI	Large bladder (>500 mL) Irregular bladder wall, Trabeculations		Recent TURBT (second-look)	
3	Obese BMI > 30 Pelvic Radiation Any open bladder surgery Bleeding disorder or Coumadin or Anti-aggregant	Urethral stricture High or narrow bladder neck Large Median lobe Large prostate (60–90 mL) Small bladder (100–250 mL) Female prolapse or cystocele	4–10	3–5 cm Large micropapillary area or suspicious for CIS (>5 cm^2^)	Prostatic urethra Bladder neck Lateral wall
4	ASA class 4–5	Not amenable to lithotomy position Very small bladder (<100 mL) Very large prostate (>90 mL) Bladder hernia Thin bladder wall	>10	>5 cm	
4.5					Posterior or Anterior wall Ureteric orifice
6					Dome Anticipate obturator jerk Diverticulum

Abbreviations: UTI: urinary tract infection.

**Table 4 cancers-12-03063-t004:** Panel participants’ characteristics and experience in urology.

Expert	Country	Age	Urology * (Years)	Oncology * (Years)	FEBU	PhD	Head of Urology **	National Association of Urology	European Association of Urology
1	F	36	4	2	-	-	0	Member NMIBC guidelines panel	Member
2	F	38	5	3	Yes	Yes	-	Board member NMIBC guidelines panel	Chairman YAU Board member YOU & ESOU
3	CZ	39	14	-	Yes	Yes	-	Member	Member
4	D	45	19	14	Yes	Yes	6	Board Member in charge of Research	Vice-Chairman NMIBC guidelines panel
5	UK	53	20	20	Yes	-	0	Member	Member NMIBC guidelines panel
6	F	58	26	26	-	Yes	-	Member	Board Member ESOU
7	CZ	58	27	22	-	Yes	10	President of National Urological Society	Chairman NMIBC guidelines panel Member Education office of the ESU
8	F	59	26	25	Yes	Yes	5	Member	EAU Board Member ESU Member
9	E	61	33	20	Yes	Yes	2	Member	EAU Board member Director of ESU NMIBC Guidelines panel
10	NL	62	28	28	-	Yes	22	Chairman bladder cancer guidelines office	Chairman MIBC guidelines panel, ESU Member

* Years since board certification, that is, 202 years of combined experience in urology and 160 years in oncology. ** Years since head of department or unit. FEBU: Fellow of the European Board of Urology, ESU European School of Urology, YAU: Young Academic Urologists, ESOU: European Society of Oncologic Urology, NMIBC: non-muscle invasive bladder cancer, MIBC: muscle-invasive bladder cancer.

**Table 5 cancers-12-03063-t005:** Questions and Likert scores for complexity and patient and tumour’s characteristics and surgical environment.

Domains	Question	Likert Scores
Patient and tumour and bladder characteristics	How likely is this characteristic to negatively impact TURBT, that is, to result in incomplete resection or prolonged surgery (>1 h) or significant intra- or postoperative complications (Clavien-Dindo Grade III and higher)?	(1) It is very unlikely to impact TURBT
		(2) It is unlikely to impact TURBT
		(3) It may occasionally impact TURBT
		(4) It is likely to impact TURBT
		(5) It is very likely to impact TURBT
Surgical Environment	How likely is the following element of the surgical environment to influence the risk of TURBT resulting in either three situations, i.e., incomplete resection according to the operator, or prolonged surgery (>1 h) or significant intra- (bleeding that requires transfusion, laparotomy) or postoperative complications (Clavien-Dindo Grade III and higher)?	(1) It is very likely to reduce the risk
		(2) It is likely to reduce the risk
		(3) It is not expected to influence the risk in either way
		(4) It is likely to increase the risk
		(5) It is very likely to increase the risk
Clinical scenarios	In the following scenario, will TURBT result in incomplete resection or prolonged surgery (>1 h) or significant intra- or postoperative complications (Clavien-Dindo Grade III and higher)?	(1) This is very unlikely to happen
		(2) This is unlikely to happen
		(3) This may occasionally happen
		(4) This is likely to happen
		(5) This is very likely to happen

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
