# Peer review of "Consensus Definition and Prediction of Complexity in Transurethral Resection or Bladder Endoscopic Dissection of Bladder Tumours"

_cancers, 2020, doi:10.3390/cancers12103063_

Round 1
Reviewer 1 Report
I think that although subjective, that the generation of a Bladder Complexity Checklist is a useful tool for clinicians.
Author Response
Dear Sir,
Many thanks for this positive review.
This research is based on experience that is the personal understanding of a complex question driven by a large set of events. Although by construct it might be considered "subjective", we believe that it should rather be understood as the product of the cumulated experience of a select set of physicians.
In that line as detailed in the Discussion section, experience can alternately be understood as a limitation or a strength.
As suggested, the manuscript was reviewed by a native English speaker (HM) for any inconsistencies in grammar or construction.
Yours Sincerely,
Bernard Malavaud
Reviewer 2 Report
It was a great honor for me to review the present paper. The authors newly developed Bladder Complexity Checklist to asses the complexity of TUR preoperatively.
1. Who developed 150 clinical scenarios? The authors should clarify the role of each author.
2. Because their study was quite interesting, the authors should provide 150 clinical scenarios in another supplementary file. As they mentioned, their research methods would be amenable to other procedures.
Author Response
Dear Sir,
The random function of the Excel software was used to produce 250 scenarios that I, as senior author, checked for clinical consistency (now introduced in the text as "by a senior author (BM)"). 150 were then drawn randomly from the set of validated scenarios to research the discrimination of individual items in the prediction of complexity.
The scenarios in the Excel format are ill adapted to the visual constraints of online publication. They were not included in the supplementary material section of the manuscrit although we would be glad to forward the reviewer the native Excel file, should it be required.
We concur that the present methodology would be readily amenable to other complex procedures that are influenced by a large number of patients' or tumors' characteristics or concurrent pathologies, such as radical prostatectomy or kidney transplant.
Many thanks for your positive appraisal of our work.
Yours Sincerely,
Bernard Malavaud